# Colorectal Cancer and Onset of Anxiety and Depression: A Systematic Review and Meta-Analysis

**Vicki Cheng** [1,2], **Niki Oveisi** [1,2], **Helen McTaggart-Cowan** [3,4], **Jonathan M. Loree** [5,6], **Rachel A. Murphy** [3,7] **and Mary A. De Vera** [1,2,8,*]

1   Faculty of Pharmaceutical Sciences, University of British Columbia, Vancouver, BC V6T 1Z3, Canada
2   Collaboration for Outcomes Research and Evaluation, Vancouver, BC V6T 1Z3, Canada
3   Cancer Control Research, BC Cancer, Vancouver, BC V5Z 1L3, Canada
4   Faculty of Health Sciences, Simon Fraser University, Burnaby, BC V5A 1S6, Canada
5   Medical Oncology, BC Cancer, Vancouver, BC V5Z 4E6, Canada
6   Division of Medical Oncology, Department of Medicine, Faculty of Medicine, University of British Columbia, Vancouver, BC V6T 1Z3, Canada
7   School of Population and Public Health, University of British Columbia, Vancouver, BC V6T 1Z3, Canada
8   Centre for Health Evaluation and Outcome Sciences, University of British Columbia, Vancouver, BC V6Z IY6, Canada
*   Correspondence: mdevera@mail.ubc.ca; Tel.: +1-604-221-8767

**Abstract:** Research suggests that colorectal cancer (CRC) is associated with mental health disorders, primarily anxiety and depression. To synthesize this evidence, we conducted a systematic review and meta-analysis of studies evaluating the onset of anxiety and depression among patients with CRC. We searched EMBASE and Medline from inception to June 2022. We included original, peer-reviewed studies that: used an epidemiologic design; included patients with CRC and a comparator group of individuals without cancer; and evaluated anxiety and depression as outcomes. We used random effects models to obtain pooled measures of associations. Quality assessment was completed using the Newcastle-Ottawa scale. Of 7326 articles identified, 8 were eligible; of which 6 assessed anxiety and depression and 2 assessed depression only. Meta-analyses showed a non-significant association between CRC and anxiety (pooled HR 1.67; 95% CI 0.88 to 3.17) and a significant association between CRC and depression (pooled HR 1.78; 95% CI 1.23 to 2.57). Predictors of anxiety and depression among patients with CRC included clinical characteristics (e.g., comorbidities, cancer stage, cancer site), cancer treatment (e.g., radiotherapy, chemotherapy, colostomy), and sociodemographic characteristics (e.g., age, sex). The impacts of anxiety and depression in patients with CRC included increased mortality and decreased quality of life. Altogether, our systematic review and meta-analysis quantified the risks and impacts of CRC on anxiety and depression, particularly an increased risk of depression after CRC diagnosis. Findings provide support for oncologic care that encompasses mental health supports for patients with CRC.

**Keywords:** colorectal cancer; mental health; anxiety; depression

## 1. Introduction

Colorectal cancer (CRC) is the third most common malignancy in both males and females and the second most common cause of cancer-related death in the world [1]. In 2020, the International Agency for Research on Cancer estimated 2 million new CRC cases and 1 million CRC deaths worldwide, representing 10% of the global cancer incidence and 9.4% of all cancer-caused mortality [2]. The negative physical impacts of CRC and its treatment are devastating to patients. As compared to the general population, patients with a CRC diagnosis report increased limitations of physical functioning, such as the inability to do housework, walk a half mile, or walk up and down stairs [3].

Altogether, the physical and functional impacts of CRC may further result in detrimental psychological impacts on a patients' overall health and quality of life [4]. In 2019, Peng et al. [5] published a literature review of 15 studies and reported that the prevalence of anxiety and depression among patients diagnosed with CRC ranged from 1.0% to 47.2% and 1.6% to 57%, respectively. However, unclear from this review was the onset of these mental health disorders, that is, whether they were already present at time of diagnosis or presented thereafter. Of particular interest are studies evaluating the onset of mental health disorders following a diagnosis of CRC. A US cohort study in 2019 used the Utah Population Database to show that patients with a CRC diagnosis were at an increased risk of any mental health disorders after diagnosis at 0–2 years (adjusted hazard ratio (aHR) 3.70; 95% confidence interval (CI) 3.47 to 3.95), >2–5 years (aHR 1.23; 95% CI 1.09 to 1.38), and ≥5 years (aHR 1.20; 95% CI 1.07 to 1.36) [6]. In addition, a 2021 cohort study in Denmark found that, compared to the cancer-free population, patients with CRC also had a significantly higher risk of depression, even 5 years after enrollment into the study (aHR 2.65; 95% CI 1.61 to 4.36) [7]. Recently, a Canadian cohort study in 2022 using administrative health databases evaluated risks of mental health disorders among patients with CRC stratified by sex and found that the risk of depression among males (aHR 1.11; 95% CI 1.06 to 1.16) and the risk of anxiety among both males (aHR 1.15; 95% CI 1.06 to 1.25) and females (aHR 1.09; 95% CI 1.02 to 1.16) were higher for those with CRC compared to the cancer-free group [8].

To our knowledge, this growing literature on mental health disorders following diagnosis of CRC has not been synthesized to date. Information across multiple studies (versus a single study) will provide better understanding on the relationship between CRC diagnosis and the onset of mental health disorders, including quantifying the association and identifying determinants of these outcomes as well as impacts on downstream patient outcomes. Furthermore, as awareness for the psycho-oncologic impacts of CRC continues to expand, comprehensive evidence on mental health outcomes will have implications for mental health care (e.g., monitoring for mental health disorders, identifying patients at risk) and advocacy for supports for patients. As such, we conducted a systematic review and meta-analysis to (1) synthesize the association; (2) identify predictors; and (3) examine impacts of anxiety and depression among patients with CRC.

## 2. Methods

### 2.1. Literature Search Strategy

Our systematic review was guided by the Preferred Reporting Items for Systematic Review and Meta-Analysis Protocols (PRISMA-P) 2020 guidelines, and completed with the PRISMA checklist (Supplementary File S1) [9]. The review was not registered; however, the protocol is available upon request. We developed a literature search strategy to identify peer-reviewed, published manuscripts relating to mental health disorders, particularly anxiety and depression, after a CRC diagnosis (Supplementary Files S1 and S2). We conducted a literature search of Ovid EMBASE and Ovid MEDLINE (R) and Epub Ahead of Print, In-Process, In-Data-Review & Other Non-Indexed Citations, Daily and Versions, from inception to 1 March 2022, and then updated the search on 28 June 2022. Our search strategies used a combination of database-dependent subject headings (e.g., Medical Subject Headings in Medline) and keywords mapping to the following concepts: CRC (e.g., "colon cancer" OR "rectum cancer" OR "colon tumor", etc.) and mental health outcomes (e.g., "depression" OR "anxiety" OR "mood disorder", etc.). Limits were added to the search to restrict results to human studies.

### 2.2. Study Screening and Eligibility Criteria

Search results were uploaded onto Covidence [10], where duplicates were removed automatically. Authors (VC and MDV) reviewed records at level one (title and abstract) and included peer-reviewed full-length studies that fulfilled all of the following criteria: (1) original study using an epidemiologic study design; (2) included individuals diag-

nosed with CRC at any age and a comparator group of individuals without cancer; and (3) evaluated mental health disorders, namely anxiety and depression, following a diagnosis of CRC. No restrictions were placed on geography, language, or availability of the full text.

### 2.3. Data Extraction and Quality Assessment

General information extracted from the included studies were: study characteristics (publication year, country, study design, data source, sample size, follow-up timeline, gender) and CRC information (age range, cancer site, cancer stage). Studies that reported mean ages with standard deviation were pooled using StataSE 17 [11]. We primarily extracted information on mental health disorders including: (1) specific condition (e.g., anxiety, depression); (2) methods used to assess/identify (e.g., International Classification of Diseases (ICD) codes, validated questionnaires such as Hospital Anxiety and Depression Scale (HADS)); and (3) reported measures of frequency (e.g., proportions) and/or association (e.g., crude and/or adjusted odds ratios (aOR), crude and/or aHR). With respect to the latter, if not reported, we made calculations based on available information, where possible. While we were primarily interested in anxiety and depression, we also extracted information on other reported mental health disorders, where relevant.

We also extracted information on predictors of anxiety and depression, which we define as factors reported to be associated with these outcomes in multivariable regression models. A priori, we anticipated potential predictors to include sociodemographic characteristics, particularly sex, based on previously reported relationships with mental health [12,13]. We also extracted information on impacts of anxiety and depression among patients with CRC, which we define as downstream patient outcomes evaluated in multivariable regression models.

We assessed the quality of included articles using the Newcastle-Ottawa Scale (NOS) [14]. For cohort and case control studies, the following score breakdown was adapted from McPheeters et al. [15]: (1) "Good" quality (possible points range: 6–8); (2) "Fair" quality (possible points range: 3–5); and (3) "Poor" quality: (possible points range: 0–2). In all of our assessments, we considered 'CRC' as the exposure variable and 'anxiety and/or 'depression' as the outcome variable. Authors (VC and MDV) first assessed and scored all the included articles independently. Differences in quality assessment scores were discussed and a final consensus score was determined.

### 2.4. Synthesis and Analysis

We conducted a narrative synthesis of findings of included studies. We used random effect models (DerSimonian and Laird) to conduct a meta-analysis and pooled reported (or calculated) measures of association for anxiety and depression among patients with CRC. Further, we used the $I^2$ test as a measure of heterogeneity, with (1) 0–40% indicating little to no heterogeneity; (2) 30–60% representing moderate heterogeneity; (3) 50–90% indicative of substantial heterogeneity; and (4) $\geq$75% describing considerable heterogeneity [16]. Forest plots were constructed for all pooled analyses, as well as funnel plots to assess publication bias.

## 3. Results

### 3.1. Search Results

As shown in Figure 1, our search strategy resulted in a total of 7326 original studies. After the removal of duplicates and conducting the title and abstract screening, 168 articles were considered during full-text review. The main reasons for excluding articles in full-text screening were: a lack of a comparator group ($n = 80$); study designs that did not match inclusion criteria ($n = 35$); and outcomes that did not match inclusion criteria ($n = 27$). Altogether, we included a total of eight studies in the systematic review.

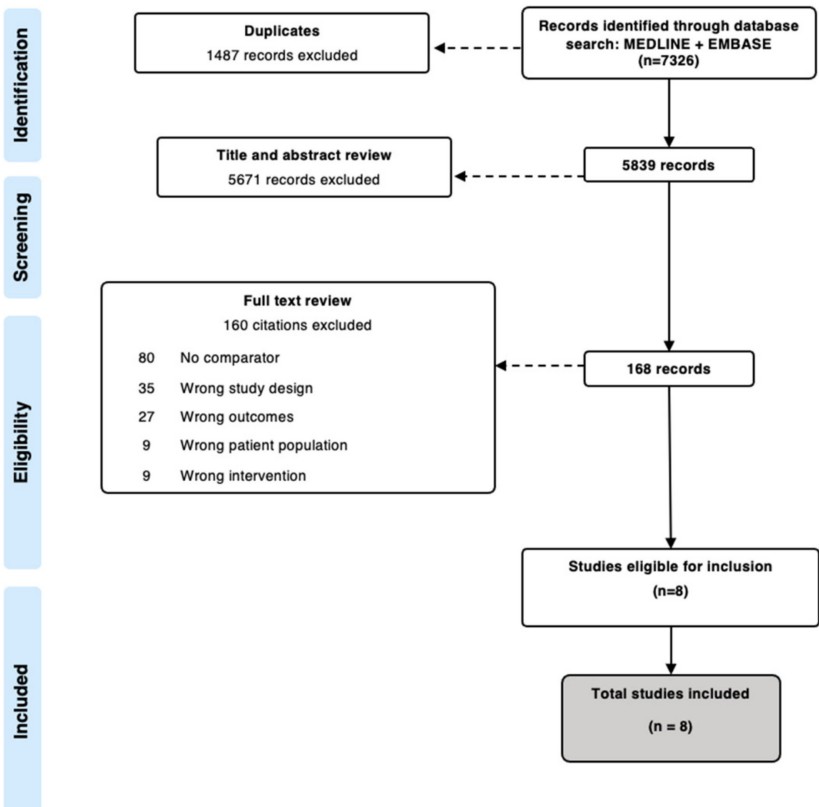

**Figure 1.** PRISMA.

### 3.2. Study Characteristics

The characteristics of the studies included are summarized in Table 1. All eight studies used a cohort study design. Out of the included studies, six studies [6,8,17–20] assessed both anxiety and depression, and two studies [7,21] assessed depression only. Included studies were predominantly conducted in developed countries: United States (*n* = 3) [6,17,21], Taiwan (*n* = 2) [18,20], Netherlands (*n* = 1) [19], Denmark (*n* = 1) [7], and Canada (*n* = 1) [8]. All studies used administrative health data from their respective countries as their main data source. The follow-up for studies ranged from 4 to 32 years. With respect to participants' sex, the proportion of male patients with CRC ranged from 47% [17] to 68% [21]. However, we note that, although studies were assessed to be reporting sex, as generally administrative health data do not capture gender, sex and gender terminology were conflated in half of the studies (*n* = 2) [17,21], where the terms 'men' and 'women' (which refer to sex) were used interchangeably with 'males' and 'females' (which refer to gender), respectively. In regard to age, the majority of studies (*n* = 7) [6,8,17–21] reported the age at the time of CRC diagnosis and one study [7] reported age at the time of the study. However, all studies varied in their reporting of age, as we observed either the reporting of mean and standard deviation or proportion according to varying age categories. The pooled age at CRC diagnosis was 68.2 years (95% CI 65.54 to 70.85). Regarding cancer stage at diagnosis, two studies [18,20] did not report cancer stage, three studies [6,7,19] included CRC patients with stage 1 to IV disease, two studies [8,17] included CRC patients with stage 0 to IV disease, and one study [21] included CRC patients with stage 1 to III disease. Lastly, the quality assessment scores of included studies resulted in a "Fair" ranking for two studies [17,19], with scores ranging from 3 to 5, and a "Good" ranking for six studies [6–8,18,20,21], with scores ranging from 6 to 8.

**Table 1.** Characteristics of included studies on mental health disorders (anxiety and depression) in CRC patients.

| Author, Year | Country | Study Design | Data Source | Follow-Up Timeline | Sample Size | | Sex (% Males) | CRC Information | | | Quality Assessment |
|---|---|---|---|---|---|---|---|---|---|---|---|
| | | | | | CRC | Non-CRC | | Age at Diagnosis | Cancer Site | Cancer Stage | |
| Zhang [17], 2010 | US | Cohort | SEER [a] Medicare | 1998–2002 | 56,182 | 265,382 | CRC: 47 / Non-CRC: 37 | 65–74: 42.7% 75–84: 41% ≥85: 16.3% | Colon, rectum | 0–IV | 4 |
| Sun [18], 2017 | Taiwan | Cohort | NHIRD [b] | 2000–2010 | 27,242 | 10,8046 | CRC: 61 / Non-CRC: 61 | 64.2 (13.5) | Colon, rectum | Not reported | 8 |
| Mols [19], 2018 | Netherlands | Cohort | Netherlands Cancer Registry | 2000–2009 | 2625 | 315 | CRC: 55 / Non-CRC: 55 | 69.4 (9.5) | Not reported | I–IV | 5 |
| Lloyd [6], 2019 | US | Cohort | Utah Cancer Registry | 1997–2013 | 8961 | 35,897 | CRC: 52 / Non-CRC: 51 | <40: 5.10% 40–49: 8.50% 50–59: 20.9% 60–69: 24.0% 70–79: 23.7% 80+: 17.8% | Colon, rectum | I–IV | 8 |
| Kjaer [7], 2021 | Denmark | Cohort | Danish Cancer Registry | 2001–2016 | 1324 | 6620 | CRC: 58 / Non-CRC: 58 | 71.3 (5.98) [c] | Colon, rectum | I–IV | 8 |
| Lee [20], 2021 | Taiwan | Cohort | NHIRD [b] | 2000–2011 | 30,391 | 30,391 | CRC: 51 / Non-CRC: 51 | 0–20: 0.3% 21–40: 8.6% 41–60: 42.6% 61–80: 42.1% >80: 6.5% | Not reported | Not reported | 7 |
| Weissman [21], 2021 | US | Cohort | IBM Explorys Database | 1999–2021 | 46,710 | N/A | CRC: 68 / Non-CRC: N/A | 18–65: 42% >65: 57% | Not reported | I–III | 6 |
| Howren [8], 2022 | Canada | Cohort | Population Data BC, BC Cancer Registry | 1985–2017 | 54.634 | 546,340 | CRC: 53.5 / Non-CRC: 53.5 | 67.6 (11.9) | Colon, rectum | 0–IV | 8 |

[a] Surveillance Epidemiology and End-Results (SEER); [b] The Taiwan National Health Insurance Research Database (NHIRD); [c] Age at the time of study.

### 3.3. Anxiety and Depression among Patients with CRC

Among included studies, six [6,8,17–20] assessed both anxiety and depression and two assessed depression only [7,21]. The majority of studies (*n* = 7) [6–8,17,18,20,21] assessed anxiety and depression using ICD-9 (International Classification of Diseases, Ninth Revision) codes, whereas only one study, Mols et al. [19], used validated questionnaires (Hospital Anxiety and Depression Scale (HADS)). Out of the studies that used ICD-9 codes, Zhang et al. [17] assessed diagnostic rates of anxiety and depression versus other studies that evaluated anxiety and depression as outcomes. Supplementary Table S3 summarizes anxiety and/or depression assessment methods and, when available, case definitions used in included studies. Of note, there were two included studies evaluating other mental health disorders [6,18]. Specifically, Lloyd et al. [6] and Sun et al. [18] both evaluated bipolar disorder using ICD-9 codes, and Lloyd et al. [6] further examined specific mental health disorders, including schizophrenia and other psychotic disorders and alcohol/substance-related disorders.

Among studies that evaluated anxiety, point estimates for reported measures of association consistently suggested increased risk, ranging from 1.19 (95% CI 1.13 to 1.25) [8] to 3.50 (95% CI 3.33 to 3.68) [18]. However, an exception was Zhang et al.'s [17] study, which, as described previously, evaluated diagnostic rates of mental health disorders and reported a hazard ratio of 0.77 (95% CI 0.71 to 0.84) [17]. Meta-analysis was feasible for five studies [6,8,17,18,20] and resulted in a pooled HR of 1.43 (95% CI 0.79 to 2.57) (Figure 2A). However, there is evidence of heterogeneity across the five studies (I² = 99.72%). Sensitivity analysis by removing Zhang et al.'s [17] study resulted in a pooled HR of 1.67 (95% CI 0.88 to 3.17) and an I² of 99.72%, still indicating the presence of heterogeneity (Figure 2B).

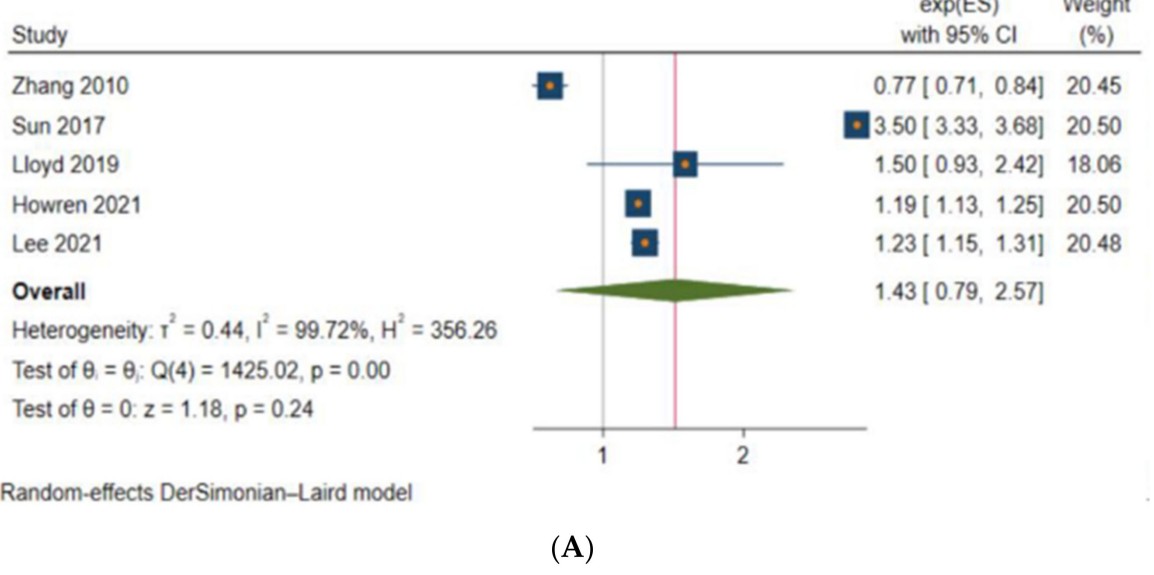

(A)

**Figure 2.** *Cont.*

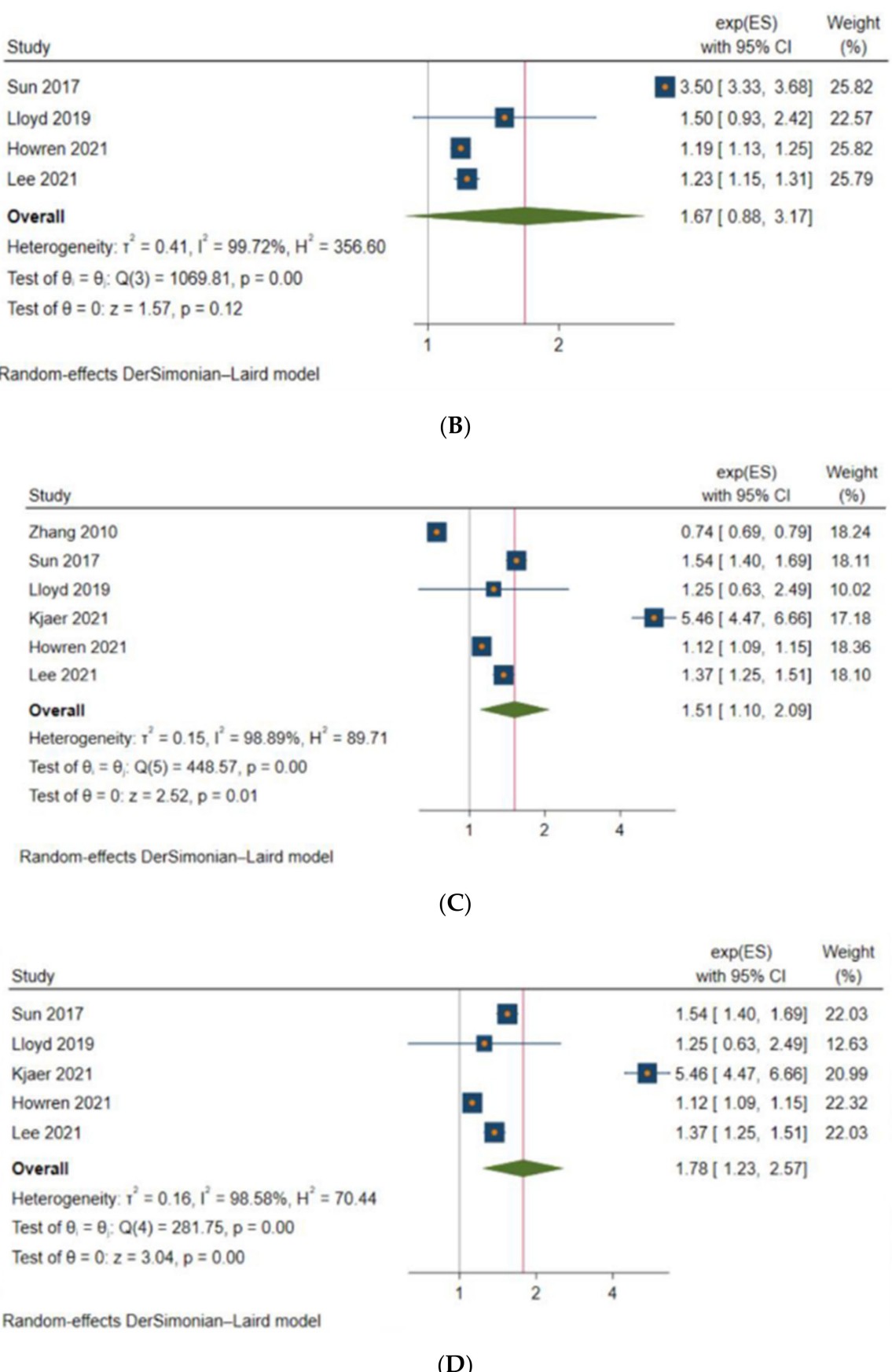

**Figure 2.** Pooled hazard ratios for anxiety and depression in patients with CRC. (**A**) Anxiety (*n* = 5); (**B**) Anxiety (*n* = 5); (**C**) Depression (*n* = 6); (**D**) Depression (*n* = 5).

Among studies that evaluated depression, point estimates for reported measures of association also consistently suggested increased risk, ranging from 1.12 (95% CI 1.09 to 1.15) [8] to 5.46 (95% CI 4.47 to 6.66) [7]. The exception again, however, was the aforementioned study by Zhang et al. that evaluated diagnostic rates of mental health disorders [17] and reported a hazard ratio of 0.74 (95% CI 0.69 to 0.79). Six studies [6–8,17,18,20] were pooled in a meta-analysis and resulted in a pooled HR of 1.51 (95% CI 1.10 to 2.09) (Figure 2C), indicating a significantly higher risk of depression in those with CRC. Of note, there was also evidence of heterogeneity across the six studies ($I^2$ = 98.89%). A sensitivity analysis was also performed by removing the study of Zhang et al. [17], which showed a stronger effect as the resulted pooled HR was 1.78 (95% CI 1.23 to 2.57), with $I^2$ = 98.58%.

### 3.4. Predictors of Anxiety and Depression

Also of interest are predictors of mental health disorders, with four studies [6,8,18,19] reporting for anxiety and five studies [6–8,18,19] for depression, as summarized in Table 2. We grouped predictors according to three categories: clinical characteristics (e.g., comorbidities, cancer stage, cancer site), cancer treatments (e.g., radiotherapy, chemotherapy, colostomy), and sociodemographic characteristics (e.g., age, sex, race). With respect to clinical characteristics, Mols et al. [19] and Lloyd et al.'s [6] analyses suggest that CRC patients with ≥1 comorbidities both during and over five years post-CRC diagnosis are associated with anxiety, with an OR of 2.86 (95% CI 1.75 to 4.55) and an aHR of 1.67 (95% CI 1.39 to 2.00), respectively. In terms of cancer stage, both Lloyd et al. [6] and Kjaer et al. [7] showed that, among CRC patients with a stage II diagnosis or higher, there is a positive association with experiencing anxiety and depression even up to five years post-CRC diagnosis [6]. It is evident that, among CRC patients, having a CRC diagnosis regardless of the site (proximal colon, distal colon, rectum) has an increased association with anxiety and depression. With respect to cancer treatments, three studies showed that CRC patients who underwent radiotherapy, chemotherapy, and colostomy exhibited a significant association of experiencing anxiety and depression [6,7,18]. Among patients with CRC who received chemotherapy treatment, Lloyd et al. [6] reported a positive association with anxiety (aHR 4.36; 95% CI 1.08 to 17.63) and depression (aHR 6.58; 95% CI 2.41 to 17.94) even up to five years post-CRC diagnosis. Lastly, sociodemographic characteristics, particularly age and sex, were also reported as significant predictors of anxiety and/or depression among patients with CRC; though findings about age were inconsistent across studies. Lloyd et al. reported that CRC patients who are ≥65 years old are more associated with experiencing anxiety (aHR 2.23; 95% 1.87 to 2.67) and depression (aHR 2.16; 95% CI 1.81 to 2.57) compared to CRC patients < 65 years old [6]. Notably, Howren et al. was the only study that showed a positive association of depression (aHR: 1.56; 95% CI 1.13 to 2.14) in the younger CRC age group (<50 years) compared with the older CRC age group (≥50 years old). With respect to sex among CRC patients, it was noted from Mols et al. and Lloyd et al. that, when compared to males, females were more likely to experience anxiety and depression. Lloyd et al. further showed that the positive association of depression in females within the CRC population persisted even after five years post-CRC diagnosis (HR: 1.15; 95% CI 0.92 to 1.44).

**Table 2.** Predictors of anxiety and depression among patients with CRC.

| Predictors | Anxiety Estimate (95% Confidence Interval) | Depression Estimate (95% Confidence Interval) |
|---|---|---|
| **Clinical characteristics** | | |
| Comorbidities | | |
| [a] Mols [19], 2018 | 2 vs. 0: OR: 2.86 (1.75–4.55) 2 vs. 1: OR: 2.56 (1.69–4.00) | 2 vs. 0: OR: 4.76 (2.70–8.33) 2 vs. 1: OR: 2.94 (1.89–4.76) |
| [b] Lloyd [6], 2019 | 0–2 years post-diagnosis 1 + vs. 0: (a)HR: 1.10 (0.99–1.22) >2–5 years post-diagnosis 1 + vs. 0: (a)HR: 1.23 (1.01–1.49) >5 years post-diagnosis 1 + vs. 0: (a)HR: 1.67 (1.39–2.00) | 0–2 years post-diagnosis 1 + vs. 0: (a)HR: 1.53 (1.31–1.79) >2–5 years post-diagnosis 1 + vs. 0: (a)HR: 1.12 (0.88–1.44) >5 years post-diagnosis 1 + vs. 0: (a)HR: 1.51 (1.19–1.91) |
| [c] Kjaer [7], 2021 | | 1–2 vs. 0: (a)HR: 1.74 (1.24–2.43) 3 + vs. 0: (a)HR: 2.74 (1.84–4.09) |
| Cancer stage | | |
| Mols [19], 2018 | IV vs. I: OR: 1.14 (0.35–3.70) IV vs. II: OR: 0.98 (0.31–3.13) IV vs. III: OR: 0.90 (0.30–2.70) | IV vs. I: OR: 2.38 (0.87–6.25) IV vs. II: OR: 2.33 (0.88–6.25) IV vs. III: OR: 1.96 (0.78–5.00) |
| Lloyd [6], 2019 | 0–2 years post-dx II vs. I: (a)HR: 1.31 (1.13–1.51) III vs. I: (a)HR: 1.34 (1.17–1.53) IV vs. I: (a)HR: 1.71 (1.48–1.99) >2–5 years post-dx II vs. I: (a)HR: 1.15 (0.90–1.48) III vs. I: (a)HR: 1.38 (1.09–1.74) IV vs. I: (a)HR: 2.63 (1.84–3.74) >5 years post-dx II vs. I: (a)HR: 0.99 (0.81–1.21) III vs. I: (a)HR: 0.86 (0.69–1.07) IV vs. I: (a)HR: 1.03 (0.58–1.85) | 0–2 years post-diagnosis II vs. I: (a)HR: 1.38 (1.11–1.72) III vs. I: (a)HR: 1.35 (1.10–1.67) IV vs. I: (a)HR: 1.68 (1.32–2.13) >2–5 years post-diagnosis II vs. I: (a)HR: 1.02 (0.74–1.41) III vs. I: (a)HR: 1.39 (1.03–1.87) IV vs. I: (a)HR: 2.59 (1.68–3.98) >5 years post-diagnosis II vs. I: (a)HR: 1.03 (0.79–1.35) III vs. I: (a)HR: 1.08 (0.81–1.43) IV vs. I: (a)HR: 0.92 (0.43–1.98) |
| Kjaer [7], 2021 | | II vs. I: (a)HR: 0.88 (0.56–1.38) III vs. I: (a)HR: 1.10 (0.70–1.73) IV vs. I: (a)HR: 3.07 (1.95–4.83) |
| Cancer site | | |
| Proximal colon | | |
| Lloyd [6], 2019 | 0–2 years post-dx (a)HR: 1.10 (0.86–1.14) >2–5 years post-dx: (a)HR: 1.76 (1.12–2.76) >5 years post-dx: (a)HR: 0.79 (0.48–1.30) | 0–2 years post-dx: (a)HR: 1.45 (1.03–2.06) >2–5 years post-dx: (a)HR: 2.18 (1.24–3.84) >5 years post-dx: (a)HR: 0.83 (0.44–1.56) |
| Rectum | | |
| Lloyd [6], 2019 | 0–2 years post-dx: (a)HR: 1.08 (0.93–1.27) >2–5 years post-dx: (a)HR: 1.43 (1.04–1.96) >5 years post-dx: (a)HR: 0.91 (0.7–1.18) | 0–2 years post-dx: (a)HR: 1.09 (0.85–1.39) >2–5 years post-dx: (a)HR: 1.52 (0.99–2.34) >5 years post-dx: (a)HR: 0.91 (0.64–1.29) |
| Kjaer [7], 2021 | | (a)HR: 0.90 (0.65–1.23) |

**Table 2.** *Cont.*

| Predictors | Anxiety<br>Estimate (95% Confidence Interval) | Depression<br>Estimate (95% Confidence Interval) |
|---|---|---|
| **Cancer treatments** | | |
| *Radiotherapy* | | |
| Sun [18], 2017 | (a)HR: 0.77 (0.69–0.86) | (a)HR: 0.83 (0.66–1.03) |
| Mols [19], 2018 | YES vs. NO: OR: 1.03 (0.70–1.52) | YES vs. NO: OR: 1.23 (0.83–1.85) |
| Lloyd [6], 2019 | 0–2 years post-dx: (a)HR: 2.48 (1.03–5.98) | 0–2 years post-dx: (a)HR: 2.74 (0.68–11.02) |
| Kjaer [7], 2021 | | (a)HR: 2.76 (1.82–4.19) |
| *Chemotherapy* | | |
| Sun [18], 2017 | (a)HR: 0.78 (0.71–0.85) | (a)HR: 1.00 (0.83–1.20) |
| Mols [19], 2018 | YES vs. NO: OR: 0.96 (0.58–1.59) | YES vs. NO: OR: 0.98 (0.58–1.67) |
| Lloyd [6], 2019 | 0–2 years post-dx<br>(a)HR: 2.28 (1.70–3.06)<br>>2–5 years post-dx<br>(a)HR: 4.36 (1.08–17.63) | 0–2 years post-dx<br>(a)HR: 2.04 (1.27–3.29)<br>>2–5 years post-dx<br>(a)HR: 6.58 (2.41–17.94) |
| Kjaer [7], 2021 | | (a)HR: 0.98 (0.67–1.42) |
| *Colostomy* | | |
| Sun [18], 2017 | (a)HR: 0.67 (0.52–0.87) | (a)HR: 0.71 (0.42–1.21) |
| Lloyd [6], 2019 | 0–2 years post-dx<br>(a)HR: 1.80 (1.48–2.18)<br>>2–5 years post-dx<br>(a)HR: 1.54 (0.99–2.39)<br>>5 years post-dx<br>(a)HR: 1.55 (0.98–2.46) | 0–2 years post-dx<br>(a)HR: 1.99 (1.52–2.61)<br>>2–5 years post-dx<br>(a)HR: 2.23 (1.46–3.40)<br>>5 years post-dx<br>(a)HR: 1.19 (0.65–2.17) |
| Kjaer [7], 2021 | | (a)HR: 1.61 (0.88–2.96) |
| **Sociodemographic characteristics** | | |
| *Age* | | |
| Mols [19], 2018 | Old vs. Young: OR: 1.01 (0.99–1.03) | Old vs. Young: OR: 1.01 (1.00–1.03) |
| Lloyd [6], 2019 | 0–2 years post-diagnosis<br>≥65 vs. <65 years: (a)HR: 1.05 (0.95–1.17)<br>>2–5 years post-diagnosis<br>≥65 vs. <65 years: (a)HR: 1.42 (1.17–1.72)<br>>5 years post-diagnosis<br>≥65 vs. <65 years: (a)HR: 2.23 (1.87–2.67) | 0–2 years post-diagnosis<br>≥65 vs. <65 years: (a)HR: 0.94 (0.80–1.11)<br>>2–5 years post-diagnosis<br>≥65 vs. <65 years: (a)HR: 0.98 (0.77–1.25)<br>>5 years post-diagnosis<br>≥65 vs. <65 years: (a)HR: 2.16 (1.81–2.57) |
| Howren [8], 2022 | <50 vs. ≥50 years: (a)HR: 1.17 (0.79–1.75)<br>Male (<50 vs. ≥50 years): (a)HR: 0.93 (0.47–1.83)<br>Female (<50 vs. ≥50 years): (a)HR: 1.34 (0.81–2.22) | <50 vs. ≥50 years: (a)HR: 1.56 (1.13–2.14)<br>Male (<50 vs. ≥50 years): (a)HR: 1.60 (1.02–2.52)<br>Female (<50 vs. ≥50 years): (a)HR: 1.49 (0.93–2.37) |
| *Sex* | | |
| Mols [19], 2018 | Female vs. Male: OR: 1.92 (1.33–2.78) | Female vs. Male: OR: 0.97 (0.66–1.41) |
| Lloyd [6], 2019 | 0–2 years post-dx<br>Female vs. Male: HR: 0.79 (0.88–0.71)<br>>2–5 years post-dx<br>Female vs. Male: HR: 0.99 (0.82–1.20)<br>>5 years post-dx<br>Female vs. Male: HR: 0.95 (0.80–1.13) | 0–2 years post-dx<br>Female vs. Male: HR: 1.38 (1.18–1.62)<br>>2–5 years post-dx<br>Female vs. Male: HR: 1.48 (1.16–1.88)<br>>5 years post-dx<br>Female vs. Male: HR: 1.15 (0.92–1.44) |

Abbreviations: (a)HR—adjusted hazard ratio; OR—odds ratio. [a] Number of comorbidities (0 vs. 2 and 1 vs. 2);
[b] Charlson Comorbidity Index of 1+; [c] Charlson Comorbidity Index of 1–2 and 3+ (excluding cancer).

*3.5. Impacts of Anxiety and Depression*

Three studies [6,19,21] additionally reported on impacts of anxiety and/or depression, when evaluated as exposures, on outcomes among patients with CRC, including health-related quality of life (QoL) [19] and mortality [6,21]. Mols et al. [19] used the European Organization for the Research and Treatment of Cancer Quality of Life questionnaire (EORTC-QLQ-C30) [22], which is designed to assess cancer patients' overall QoL, particularly measuring patients' physical, psychological, and social functions from 0 to 100, with high global QoL scores (e.g., 100) indicating better QoL. Authors found that CRC patients who always reported symptoms of anxiety over three years had global QoL scores that were on average 20.6 to 23.2 points lower than the scores of CRC patients who did not report anxiety symptoms. Similarly, CRC patients who always reported depressive symptoms had global QoL scores that were on average 25.9 to 28.6 points lower than the scores of CRC patients who never reported depressive symptoms. As per the evidence-based interpretation guidelines for global QoL scores [23], >13 points is the threshold for a significantly large mean difference in QoL. Thus, the above differences in QoL reported from Mols et al. [19] in CRC patients with and without anxiety and depression are of clinical relevance (>15-point difference), suggesting these mental health disorders have a significant impact on a CRC patients' overall QoL. With respect to anxiety and depression and their impact on mortality, Weissman et al. [21] reported significantly higher odds of death in patients with new-onset depression after a CRC diagnosis (OR: 2.23; 95% CI: 2.02 to 2.87). Lastly, Lloyd et al. [6] found that an increased mortality is more associated with CRC patients diagnosed with any mental health disorders (HR: 2.18; 95% CI 2.02 to 2.35) and depression (HR: 2.10; 95% CI 1.92 to 2.28) than CRC patients without.

## 4. Discussion

In our systematic review and meta-analysis, we aimed to synthesize current evidence on the association, predictors, and impacts of anxiety and depression among patients with CRC. Altogether, we included eight studies that evaluated anxiety and depression among 228,069 patients with CRC. Our meta-analyses suggested that individuals with CRC have a 51% increased risk (pooled HR: 1.51; 95% CI 1.10 to 2.09) of experiencing depression after diagnosis. Findings for anxiety suggested no association as the pooled estimate of 1.43 was not statistically significant, suggesting the need for further studies. Importantly, several predictors of anxiety and depression in patients with CRC were identified including clinical characteristics (e.g., comorbidities, cancer stage, cancer site), cancer treatments (e.g., radiotherapy, chemotherapy, colostomy), and sociodemographic characteristics (e.g., age, sex). Additionally, we identified the impacts [6,19,21] of anxiety and depression on CRC patients, particularly on patient outcomes of health-related QoL and mortality. Overall, our synthesis provides a comprehensive understanding of anxiety and depression in CRC patients to date. These findings highlight that there is an overall positive association between a CRC diagnosis and experiencing these mental health disorders, particularly depression, which then have downstream effects on patient outcomes. Our findings also suggest the importance of identifying and monitoring mental health disorders during encounters of care for CRC patients with identified predictors of anxiety and/or depression as potential targets.

Previous research has shown the concern of patients experiencing anxiety and/or depression symptoms following a cancer diagnosis [24,25]; thus, our review aimed to expand on this area by highlighting the onset of predominantly discussed mental health disorders of anxiety and depression specifically after a CRC diagnosis. In regard to mental health disorders and CRC, prior to our systematic review, Peng et al.'s [5] literature review in 2019 identified 15 studies and reported that CRC patients exhibited a significant burden of anxiety and depression symptoms, with the prevalence of anxiety and depression among patients diagnosed with CRC ranging from 1.0% to 47.2% and 1.6% to 57%, respectively. Of note, Peng et al. [5] focused on studies that looked at the burden (prevalence) of anxiety and depression, where CRC patients were already experiencing mental health symptoms

at the time of their CRC diagnosis. In significant contrast, a majority of the included studies (*n* = 5) [6,7,17,18] in our systematic review were those that assessed the onset of anxiety and depression after a CRC diagnosis. Therefore, our review adds to the current literature around mental health in CRC patients as, from the included studies, we found a 51% higher risk of depression among CRC patients compared to individuals without CRC. Biological mechanisms could explain this increased risk of depression, as evidence suggests that the inflammatory nature of cancer may be closely linked to anxiety and depression [26]. Different biological processes including elevated inflammatory mediators (i.e., C-Reactive Protein (CRP)), tissue damage, and chronic stress response may predispose cancer patients to depression [26]. As highlighted by Renna et al., nearly one-third of CRC patients who reported clinically significant levels of depressive symptoms also had high CRP levels (defined as levels >3 mg/L) [27]. These biological mechanisms may continue to build up stress that goes beyond the coping mechanisms of cancer patients, which may result in patients experiencing depressive symptoms persisting even after treatment [26,28]. Moreover, having a CRC diagnosis may negatively impact an individuals' body image, possibly leading to an impairment on ones' sexual health; all of which may all contribute to mental health disorders persisting in the long term within this patient population [3,4,29].

It is important to contextualize findings with respect to anxiety and depression outcomes. Among patients with CRC, anxiety was often less evaluated (*n* = 6 studies) [6,8,17–20] compared to depression (*n* = 8 studies) [6–8,17–21], as is evident across the included studies in this systematic review. Furthermore, in our meta-analysis, we found a lack of statistical significance in the association between anxiety and CRC patients. This may be driven by the small number of studies that assessed anxiety in CRC, compared to depression, in our review. The lack of a statistically significant association between anxiety and CRC patients also reflects high $I^2$ values ($I^2$ = 99.72%, sensitivity analysis $I^2$ = 99.72%). In terms of depression, although we found a statistically significant association between depression and CRC patients, there is also a considerably high heterogeneity in the pooled HRs ($I^2$ = 98.89%, sensitivity analysis $I^2$ = 98.58%). As the results of our meta-analyses suggested heterogeneity between studies, it is plausible that the high heterogeneity may be due the variation between the countries (e.g., Eastern and Western) where our included studies were conducted. In 2020, Krendl et al. published a cross-national epidemiologic survey, revealing higher levels of stigma around mental illness among individuals in Eastern countries compared to those from Western countries [30]. The findings in Krendl et al.'s study reveal that the difference between the levels of stigma around mental disorders in Eastern compared to Western countries is largely due to cultural differences in the attributions about mental illness [30]. Correspondingly, Figure 2 shows opposing associations between two individual studies, Zhang et al. [17] and Sun et al. [18] (both studies conducted in an Eastern country, Taiwan), further suggesting that cultural differences and the potential lack of education and knowledge around what anxiety and depression are may influence patients' help-seeking behavior and affect their awareness to seek mental health support, all of which may lead to underdiagnosis of mental health disorders [31,32]. Thus, the high stigma around mental health disorders in different cultures may explain the considerable heterogeneity found in our meta-analyses, which may impact the statistically significant associations drawn from our pooled estimates in regard to the depression outcome. Moving forward, pooling findings from studies conducted from similar global regions evaluating anxiety and/or depression may reduce heterogeneity. As a result, given the considerably high heterogeneity between published anxiety and depression pooled estimates in CRC patients thus far, this prompts future research to further characterize the onset of specific mental disorders in patients with CRC.

Beyond quantifying the association of mental health disorders with a CRC diagnosis, it was also imperative to identify predictors of anxiety and depression among CRC patients to inform practical recommendations. Thus, another important contribution of our synthesis was extracting information on different predictors as these may represent potential targets for intervention or means for identifying high-risk patients. Most studies (*n* = 5) [6–8,18,19]

reported predictors of anxiety and depression, all of which showed similar patterns of a positive association between having a CRC diagnosis and possessing different clinical, treatment, and sociodemographic characteristics. Notably, studies consistently reported between a range of a two- to six-fold increased association with mental health disorders in CRC patients having ≥1 comorbidities [6,7], diagnosed with stage II or higher [6,7], and when undergoing different types of cancer treatments [6,7,33]. Findings with respect to age were conflicting across studies as Lloyd et al. [6] reported that those who are ≥65 years old are positively associated with experiencing anxiety (aHR 2.23; 95% 1.87 to 2.67) and depression (aHR 2.16; 95% CI 1.81 to 2.57). Howren et al. [8], however, showed a significant association with depression (aHR: 1.56; 95% CI 1.13 to 2.14) in the younger CRC age group (<50 years). Over the last decade, evidence has demonstrated an increase in the incidence of young-onset CRC (yCRC), which is CRC diagnosed in adults less than 50 years of age [33,34]. The incidence of yCRC in Canada has increased by a mean annual percentage change (APC) of 3.47% for women since 2006 and by 4.45% for men since 2010 [35]. As a result, the increasing risk of yCRC warrants future research to further investigate a potential age effect in a CRC diagnosis and implications for early CRC screening in the younger age group [36]. With respect to sex, the significant association between CRC females and experiencing anxiety and depression more than males aligns with current literature, where females have been shown to have a two-times higher prevalence and an increased cumulative incidence of depression compared to males [37,38]. Although having a cancer diagnosis will contribute to experiencing mental health symptoms [24], sex also plays a huge role as being a female may be a major risk factor for mental health symptoms compared to males [39,40]. Furthermore, the positive association with anxiety and depression in CRC patients with the aforementioned predictors was shown to persist even after five years post-CRC diagnosis [6]. Individuals who live with chronic medical conditions often experience chronic pain and emotional stress, both of which are associated with the development of anxiety and depression symptoms [41,42]. Additionally, a higher risk of anxious and depressive symptomatology in advanced CRC stages aligns with previous research that has shown that disease stage was directly associated with emotional distress [25,43,44]. Therefore, the associations between a CRC diagnosis and clinical, treatment, and sociodemographic characteristics point to the importance of identifying high-risk CRC patients with these predictors early on in diagnosis [24]. Following the early identification of these predictors calls for enhanced psychological interventions to integrate mental health care and patient resources to aid patients emotionally and mentally throughout different points of care.

Lastly, an important consideration of our review is the observation of mental health disorders on the impact on overall patient health outcomes. Observation of these in three studies [6,19,21] in this synthesis built on previous research showed that the onset of anxiety and/or depression is associated with downstream outcomes of increased mortality and decreased patient quality of life [44–47]. Given negative associations with physical, emotional, and cognitive functioning [19], leading to a significantly impaired quality of life [19,47], increased integration of mental health care is particularly vital in CRC management. Collectively, anxiety and depression's leading effects on downstream outcomes further justifies the strong need for early psychological intervention within a patients' CRC journey as well as continued mental health care support even after CRC treatment.

The strength and limitations of a systematic review deserve discussion. Our systematic review describing the risk of anxiety and depression in patients with CRC identified articles using a thorough search strategy. An original and updated search further ensured the comprehensive and timely capture of relevant studies to date. To our knowledge, this is the first study aiming to synthesize and critically appraise evidence from observational studies analyzing the association of anxiety and depression following a diagnosis of CRC. As in any other systematic review, the inclusion of relevant studies may have been limited by publication bias, as shown by funnel plots in Supplementary Figure S1, which is a limitation of our review. This systematic review focused on the association between a

CRC diagnosis and anxiety and depression, as these mental health outcomes are most frequently studied; however, a complete understanding of mental health comorbidities in CRC patients could also expand to include other types of mental health disorders, such as bipolar disorder. We identified that only two [6,18] of our included studies included information on bipolar disorder.

Altogether, our systematic review and meta-analysis provide a comprehensive synthesis of mental health outcomes among CRC patients. Further synthesizing the predictors and impacts of anxiety and depression provides practical information for the identification of at-risk CRC patients, as well as integrating psychological and mental health care interventions early on in CRC patients' journeys. Our findings prompt future research to further expand on the scope of anxiety and depression in individuals with a CRC diagnosis. Furthermore, our results have implications to support the need for mental health care support before, during, and after a CRC diagnosis, as well as informing future research to substantiate the risk of anxiety in CRC patients. As mental health disorders may go unnoticed in CRC clinical care, our findings further provide significance for clinical practice health care providers (including oncologists, general practitioners, pharmacists, and nurses who routinely interact with CRC patients), as they should be aware of anxiety and depression when treating patients diagnosed with CRC.

**Supplementary Materials:** The following supporting information can be downloaded at: https://www.mdpi.com/article/10.3390/curroncol29110689/s1, File S1. PRISMA 2020 Checklist; Table S1: EMBASE Ovid Search (1974 to March 01, 2022); updated search ran June 29, 2022; Table S2: Ovid MEDLINE(R) and Epub Ahead of Print, In-Process, In-Data-Review & Other Non-Indexed Citations, Daily and Versions(R) Search <1946 to March 01, 2022; updated search ran June 29, 2022>; Table S3: Summary of anxiety and depression assessment methods and definitions; Figure S1: Funnel plots of included studies in anxiety and depression.

**Author Contributions:** V.C. contributed to conceptualization, investigation, data curation, methodology, project administration, formal analysis, visualization, data interpretation, and writing the original draft. V.C. and N.O. contributed to investigation, methodology, and conceptualization. H.M.-C., J.M.L., and R.A.M. contributed to conceptualization, methodology, and data interpretation. M.A.D.V. contributed to conceptualization, data curation, formal analysis, investigation, methodology, project administration, resources, supervision, visualization, data interpretation, and writing the original draft. M.A.D.V. and H.M.-C. contributed to funding acquisition. All study authors reviewed and edited the manuscript. All authors have read and agreed to the published version of the manuscript.

**Funding:** This research was funded by a Project Grant from the Canadian Institutes of Health Research, "Examining the epidemiology, treatment, and outcomes in young-onset colorectal cancer" (Funding reference number: PJT-159467). The funder had no role in study design, data collection and analysis, decision to publish, or preparation of the manuscript. De Vera holds a Tier 2 Canada Research Chair.

**Conflicts of Interest:** J.M.L has received research funding from Ipsen, Amgen and Foundation Medicine and consulting fees from Roche, Bayer, Amgen, Ipsen, Advanced Accelerator Applications and Novartis. All other authors declare that they have no conflicts of interests.

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
