# Peer review of "Colorectal Cancer and Onset of Anxiety and Depression: A Systematic Review and Meta-Analysis"

_curroncol, doi:10.3390/curroncol29110689_

Round 1

Reviewer 1 Report

The authors present a systemic review of the association of CRC diagnosis and the mental health issues of depression and anxiety. It is interesting to note the different studies that have taken on this topic around the world and the similarities of their findings. It will be useful for readers to have a central location to review these data and a very well done systemic analysis of their findings. There are minor grammatical errors to be addressed. 

Reviewer 2 Report

Thank you for submitting your systematic review to Current Oncology. My comments are as below.

Overall:

11)   Comprehensive search is essential for a systematic review. Authors only visit 2 databases for all relevant papers that could miss important evidence. The quality of the review has been affected.

2 2)     Any meta-analysis has been done for the predictors?

3 3)    The period in the literature search is limited from 20220228 to 20220629. This period is too short that will miss lot of evidence.

4 4)     The literature search was limited to embase, but the authors conducted the search using EMBASE.

5 5)     The focus of this review on anxiety and depression but the keywords used for the literature search were the key terms for psychiatric /mental problems. The relevant keywords have to be used for the relevant articles for the review. Definitely, your review is inadequate to understand anxiety and depression in people with CRC.

Reviewer 3 Report

This is a systematic review and meta analysis investigating and evaluating the onset of anxiety and depression among individuals with colorectal cancer.

Title: Title is clear and encapsulates the scope of the study appropriately.

Abstract:  Abstract is well written. If space permits, may be helpful to incorporate subtitles to organize section (i.e Introduction, Objectives, Results, Discussion, etc.)

Introduction: Introduction is very well written.

Methods: Well written and clear.

Results:  Well written. It might be helpful to write out full name of various acronyms like IDS-9 or HADS in section 3.3.

Discussion: Well written – interesting explanation on heterogeneity due to stigma.

Conclusion: More information on future scope of studies is recommended.

Overall, this paper was very well written and is an interesting read. I recommend this paper be published. 

Round 2

Reviewer 2 Report

The systematic review has been improved but only inclusion of articles in few months is limited. The authors should add this as an limitation. 
